# Hepatitis B Virus Epsilon (ε) RNA Element: Dynamic Regulator of Viral Replication and Attractive Therapeutic Target

**DOI:** 10.3390/v15091913

**Published:** 2023-09-12

**Authors:** Lukasz T. Olenginski, Solomon K. Attionu, Erica N. Henninger, Regan M. LeBlanc, Andrew P. Longhini, Theodore K. Dayie

**Affiliations:** 1Center for Biomolecular Structure and Organization, Department of Chemistry and Biochemistry, University of Maryland, College Park, MD 20742, USAreganleblanc@gmail.com (R.M.L.);; 2Department of Biochemistry, University of Colorado, Boulder, CO 80309, USA; 3Neuroscience Research Institute, University of California, Santa Barbara, Santa Barbara, CA 93106, USA; 4Department of Molecular, Cellular and Developmental Biology, University of California, Santa Barbara, Santa Barbara, CA 93106, USA

**Keywords:** HBV, RNA, structure, small molecules, therapeutics

## Abstract

Hepatitis B virus (HBV) chronically infects millions of people worldwide, which underscores the importance of discovering and designing novel anti-HBV therapeutics to complement current treatment strategies. An underexploited but attractive therapeutic target is ε, a *cis*-acting regulatory stem-loop RNA situated within the HBV pregenomic RNA (pgRNA). The binding of ε to the viral polymerase protein (P) is pivotal, as it triggers the packaging of pgRNA and P, as well as the reverse transcription of the viral genome. Consequently, small molecules capable of disrupting this interaction hold the potential to inhibit the early stages of HBV replication. The rational design of such ligands necessitates high-resolution structural information for the ε–P complex or its individual components. While these data are currently unavailable for P, our recent structural elucidation of ε through solution nuclear magnetic resonance spectroscopy marks a significant advancement in this area. In this review, we provide a brief overview of HBV replication and some of the therapeutic strategies to combat chronic HBV infection. These descriptions are intended to contextualize our recent experimental efforts to characterize ε and identify ε-targeting ligands, with the ultimate goal of developing novel anti-HBV therapeutics.

## 1. Introduction

Hepatitis B virus (HBV) is a member of the Hepadnaviral family and is the smallest DNA virus known to infect animals. The HBV genome is 3.2 kilobases (kB) [1,2,3,4,5], consisting of partially double-stranded, relaxed circular DNA (rcDNA) that is covalently attached to a multifunctional viral polymerase protein (P) (Figure 1A) [1,6,7]. P comprises four domains within a single polypeptide chain: a reverse transcriptase (RT), a middle spacer, an RNase H (RH), and a terminal protein (TP) domain [8,9,10,11,12,13]. The current Food and Drug Administration (FDA)-approved treatments for chronic HBV (cHBV) infection are interferon (IFN)-α and nucleo(t)ide RT inhibitors (NRTIs). Regrettably, these treatments are not curative and are often accompanied by off-target effects [14,15,16,17,18]. Specifically, NRTI therapy necessitates lifelong treatment and is susceptible to resistance-related mutations [14,15,16]. IFN-α treatment, on the other hand, is rife with adverse effects [17,18], with some mimicking the symptoms of cHBV infection. These limitations underscore the need for alternative anti-HBV therapeutic strategies.

One underexploited but attractive therapeutic target is the *cis*-acting RNA regulatory stem-loop known as epsilon (ε), situated at the 3′- and 5′-ends of the pregenomic RNA (pgRNA) [8,19,20,21,22,23]. HBV replication is initiated when P binds the 5′-end ε, leading to the initiation of protein-primed reverse transcription [19,24,25,26] and packaging [7,27] of P and the pgRNA into subviral core particles. Consequently, small molecules capable of disrupting this interaction, by binding to the ε–P complex or either component separately, have the potential to inhibit the early stages of HBV replication.

However, the lack of structural data for P prevents a structure-informed design of anti-HBV small molecules. Encouragingly, our recent structural elucidation of ε by solution nuclear magnetic resonance (NMR) spectroscopy [28] offers a crucial foundation for the future design of ε-targeting ligands, which is the focus of this review. Initially, we will provide a brief summary of the key events of the HBV lifecycle (Section 2) and of the existing HBV treatments (Section 3). Subsequently, we will discuss the characterization of ε (Section 4) and our efforts to identify ε-targeting ligands (Section 5). Finally, we suggest future directions to facilitate the development of ε-targeting anti-HBV therapeutic strategies (Section 6).

We do not intend for this work to be an exhaustive overview of HBV replication or the many therapeutic strategies to combat cHBV infection. We refer the reader to previous comprehensive reviews of these topics [29,30,31,32,33]. Here, our focus is on ε as a potential therapeutic target, and we include only details necessary to help contextualize recent efforts at discovering ε-targeting ligands.

**Figure 1 viruses-15-01913-f001:**
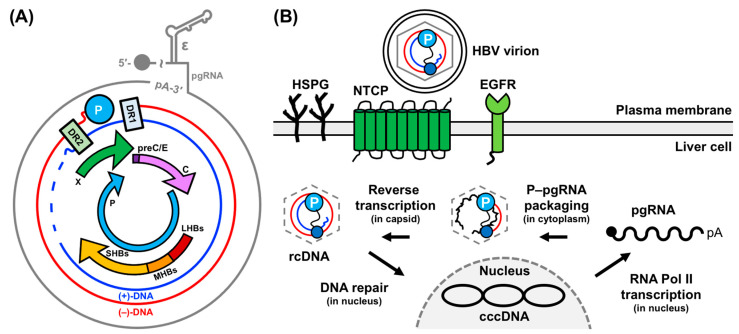
HBV genome organization and lifecycle. (**A**) Schematic of the 3.2 kB HBV rcDNA genome, depicting the pgRNA, negative (−)- and positive (+)-DNA strands, and its four open reading frames and seven gene products. Critical elements such as the attachment of P to (−)-DNA, direct repeats 1 (DR1) and 2 (DR2), and ε are highlighted. (**B**) Schematic of HBV genome replication, encompassing liver cell entry to complete genome conversion. This figure is adapted from [29,34].

## 2. HBV Replication: Molecular Mechanisms and Critical Interactions 

HBV replication (Figure 1B) begins with infectious virions binding to the sodium-taurocholate co-transporting polypeptide receptor (NTCP) [35], heparan sulfate proteoglycans (HSPG) [36,37], and/or epidermal growth factor receptor (EGFR) [38,39] of the host liver cell. Following infection, the rcDNA is imported into the host cell nucleus and repaired to form a covalently closed circular DNA (cccDNA) [1,40]. This cccDNA is transcribed into several genomic and subgenomic RNAs by the host RNA Polymerase (Pol) II and the viral transcripts are exported to the cytoplasm. Here, the pgRNA serves as the mRNA for translation into viral proteins [1,40]. Then, the pgRNA and P are selectively packaged into immature capsids, where reverse transcription regenerates new rcDNA genomes [7,8,19,20,21,22,23,27,40,41]. These mature rcDNA-containing nucleocapsids can undergo further rounds of replication to amplify the cccDNA pool or become enveloped and released from the host cell as progeny virions. Below, we delve into these genomic events, with a special emphasis on the steps involving ε.

### 2.1. Conversion of rcDNA to cccDNA to pgRNA 

The HBV genome enters the host cell as rcDNA (Figure 1B), which has many unique characteristics. For example, the outer (−)-DNA strand (i.e., with opposite polarity to mRNA) is complete, whereas the inner (+)-DNA strands (i.e., with the same polarity as mRNA) are incomplete (Figure 1A). Moreover, the 5′-end of the (−)-DNA is covalently linked to P, and the 5′-end of the (+)-DNA includes a pgRNA-derived oligonucleotide (discussed further in Section 2.2) (Figure 1A). The conversion of rcDNA to cccDNA necessitates the removal of these features and the covalent ligation of the (−)- and (+)-DNA strands. However, a detailed understanding of how this is achieved, and what regulatory factors are involved, remains elusive, owing, in part, to the difficulty of detecting cccDNA in the presence of excess rcDNA [34,42]. Potential regulatory models have been the subject of intense review [31]. Notably, it has been shown that the DNA repair enzyme tyrosyl-DNA-phosphodiesterase 2 (TDP2) cleaves P from the (−)-DNA strand of the rcDNA [43].

Once formed, the cccDNA is sufficiently stable (with half-life of ~30–60 days as measured for the related duck HBV system [44,45]) to survive cell division and persist in infected cells. The circular architecture of the cccDNA facilitates its intracellular amplification by positioning the Pol II promoter and enhancer sequences directly upstream of the genomic RNA start sites [1,2,3,4,5]. Consequently, cccDNA is continuously passed on to progeny, even during antiviral therapy [46]. Another critical role of the cccDNA is templating RNA transcription by host Pol II. The cccDNA contains seven gene products in four open reading frames: precore (preC/E), core (C), P, presurface 1 (LHBs), presurface 2 (MHBs), surface (SHBs), and X (Figure 1A) [1,47]. Each RNA features a 5′-cap and 3′-polyadenyltaed (poly-A) tail, serving as an mRNA. 

The RNA essential for HBV replication is the pgRNA, encompassing the entire genome plus a terminal redundancy of approximately 120 nucleotides (nt) that contains a second copy of the DR1 and ε elements and a poly-A tail at the 3′-end (Figure 1A) [1,2,3,4,5]. The pgRNA serves as the mRNA for C and P proteins [1] and the template for reverse transcription of new rcDNA genomes [7,8,19,20,21,22,23,27,40,41].

### 2.2. P–pgRNA Packaging and Reverse Transcription

The next critical phase in HBV genome replication is the packaging of pgRNA and P into immature capsids, followed by reverse transcription (Figure 1B). These processes require the binding of P to ε, a *cis*-acting regulatory stem-loop RNA situated at the 5′-end of the pgRNA (Figure 2A) [8,19,20,21,22,23]. A second copy of ε is also located at the 3′-end of the pgRNA but it does not bind P (Figure 2A) [1,2,3,4,5]. The interaction between the 5′-end ε and P initiates protein-primed reverse transcription [19,24,25,26] and facilitates P–pgRNA packaging into immature capsids formed by C dimers [7,27]. The priming reaction yields a 3 nt DNA motif, with its 5′-end covalently attached to a tyrosine residue (Y63) in the TP domain, templated from the 6 nt priming loop (PL) bulge within ε [9,10,11,19,24,25,26] (Figure 2A). A mechanistic understanding of the ε–P interaction requires high-resolution structural data of the complex or its individual components, which has proven difficult with traditional structural biology techniques (discussed further in Section 4.1 and Section 4.2).

### 2.3. (−)-DNA Strand Synthesis

The P−DNA complex then translocates to the 3′-proximal DR1 element where (−)-DNA strand synthesis initiates from the 3 nt DNA primer (Figure 2A) [48,49]. The initial (−)-DNA strand synthesis is templated from the 5′-UUC-3′ sequence in the ε PL. However, elongation is templated from the same sequence in the 3′-end DR1 motif nearly ~3 kB away (Figure 2B). Consequently, the TP-bound DNA must translocate a substantial distance in sequence space. Given that there are approximately 20 additional 5′-UUC-3′ motifs within the pgRNA [1], and that less than 4 nt sequence identity is required between the template and target [19], additional regulatory elements must be involved to ensure proper translocation. While the exact mechanism is unknown, one model suggests that the 3′-DR1 and 5′-ε elements are brought into close proximity via a “closed-loop” formation of the pgRNA, which is facilitated by an RNA–protein complex that links 5′-cap [50] and 3′-poly-A [51] binding proteins. The requirement of P–pgRNA packaging for close proximity of ε and the 5′-cap [50] lends some support to this hypothesis. An alternative model posits that a long-range interaction between ε and another *cis*-element known as Φ, which has partial complementarity to ε and is slightly upstream of DR1, is required for (−)-DNA synthesis [52,53,54]. In partial support of this hypothesis, mutations that impair base pairing between ε and Φ reduce (−)-DNA synthesis [55].

Regardless of the validity of these models, the translocation process must transform the ε–P complex. Initially, P facilitates protein priming with Y63 within the TP domain, then it enables DNA copying, and finally it replaces the ε template with DR1. Consequently, P must have distinct initiation and elongation modes, akin to protein-priming polymerases [56]. The product of (−)-DNA synthesis is a DNA copy of the pgRNA from its 5′-cap to the 5′-UUC-3′ motif in the 3′-DR1, including approximately 10 nt 3′- and 5′-end redundant sequences (Figure 2B). As the (−)-DNA strand is synthesized, the pgRNA is degraded by the RH domain of P, except for approximately 18 nt of its 5′-terminus, which includes the capped 5′-end DR1 (Figure 2B) [57]. This 5′-capped RNA then serves as the template for (+)-DNA strand synthesis (Figure 2C). 

### 2.4. (+)-DNA Strand Synthesis

To ensure the formation of rcDNA and not merely the production of double-stranded and linear DNA, the RNA primer must be transferred to the 3′-proximal DR2 motif (Figure 2C). This second template switch requires RNA primers with a 5′-cap and the DR1 motif. Interestingly, the template 5′-DR1 translocates to the 3′-DR2 rather than the initial 3′-DR1 motif (Figure 2C), despite having greater complementarity to the latter. As before, this observation suggests an additional level of control for efficient (+)-DNA strand synthesis, though the precise regulatory mechanism remains unknown. Nevertheless, from its new location on DR2, the RNA primer is elongated towards the TP-bound 5′-end of the (−)-DNA, including the 5′-end redundant region (Figure 2C). 

Additional elongation requires circularization, which is facilitated by a third template switch (Figure 2D). In this final transformation, the growing (+)-DNA strand is transferred from the 5′- to 3′-end redundancy on the (−)-DNA template, where its final extension yields the rcDNA (Figure 2E). While the sequence requirements of both redundant ends are critical, additional *cis*-acting elements have been hypothesized to play important roles [54,58]. Intramolecular base pairing is likely an important mechanism to ensure the proper shape and necessary contacts within the HBV genome that are needed to facilitate the three template switches that form the rcDNA [54].

### 2.5. Perspectives and Challenges to Mechanistic Studies of HBV Replication

While many of the key insights of HBV replication have been revealed using reverse genetics in transfected cells, the complex interplay between viral components and interactions with other host factors (HFs) often requires near-native experimental systems. For example, the template switches required to reform the rcDNA (Figure 2) depend on various RNA–RNA and RNA–protein interactions that only occur in the context of assembled nucleocapsids [50,51,52,53,54,55,58]. Given the lack of robust experimental systems, our understanding of cccDNA formation, initiation of reverse transcription, and (−)- and (+)-DNA strand synthesis remains incomplete. 

Nevertheless, in vitro reconstitution systems have been indispensable for mechanistic understanding, as evidenced by the discovery of reverse transcription initiation by ε [19,49] and the chaperone dependence of P (more on this in Section 4.2 and Section 4.3). However, there is still a dire need to reveal these molecular events at the atomic level with high-resolution structural biology. Ideally, we would capture the ε–P complex before and after each step of HBV genome replication (Figure 2). Given that the only relevant high-resolution structures are the HBV C protein without its nucleic acid binding domain [59] and our recent full-length ε [28], this remains an ambitious task. Nevertheless, recent advances in cryo-electron microscopy (cryo-EM) [60,61,62] may soon change this.

## 3. Tackling HBV: Insights into Viral Replication and Evolving Therapeutic Strategies

With a brief introduction to the early stages of HBV genome replication and a clear understanding of the key protein (i.e., HBV P) and RNA (i.e., ε) actors involved, we will now discuss how this information has been leveraged to develop anti-HBV therapies.

### 3.1. Global Burden of HBV

Globally, approximately one in four people have been exposed to HBV and more than 300 million people are chronically infected, leading to around 800,000 deaths annually [63,64,65]. Moreover, HBV is responsible for about 23% of all cases of cirrhosis (i.e., severe liver damage) and roughly 40% of hepatocellular carcinoma (HCC) cases [66]. Consequently, HBV infection is a significant global heath burden, especially in developing countries. According to the World Health Organization (WHO), cHBV infection is highest in the Western Pacific and Africa, where 116 and 81 million people are chronically infected, respectively (Figure 3). Even in developed countries with comprehensive vaccination programs and increased availability of treatment, the burden of HBV-related diseases remains substantial. In the United States, the prevalence of cHBV infection is estimated at around 0.27% (i.e., 0.8–1.4 million) [67], but this figure rises to 10–15% (i.e., 3–5 million) within Asian American communities [68]. In Europe, the prevalence of cHBV infection ranges from about 0.2–7%, affecting an estimated 14 million people (Figure 3). Unfortunately, up to 90% of these chronically infected individuals are unaware of their infection [64,69], suggesting that infection statistics are likely underestimates.

### 3.2. Current Treatments of cHBV Infection

The goal of treating cHBV is to prevent cirrhosis, liver failure, and HCC. Treatment end points are designed to correlate with clinical outcomes, and can be classified as biochemical, virological, serological, and histological [18]. The biochemical end point involves normalizing levels of alanine aminotransferase (ALT), whereas the virological endpoint entails suppressing cccDNA to undetectable levels [18]. In addition, the serological end point refers to the loss or seroconversion of hepatitis B e (HBeAg) and surface (HBsAg) antigens [18]. Finally, the histological endpoint involves reducing necrosis (i.e., liver tissue damage) and inflammation without increasing liver scarring [18]. Currently, there are eight FDA-approved treatments for cHBV that help patients achieve these outcomes to varying degrees.

#### 3.2.1. FDA-Approved cHBV Treatments

The eight FDA-approved treatments for cHBV infection include IFN-α and its polyethylene glycol (PEG)-modified form, along with six NRTIs (Figure 4A). As their names imply, NRTIs inhibit (−)-DNA strand elongation by the RT domain of P [14,15,16]. The mode of action of IFN-α is less clear, but it is known to possess general antiviral, immunomodulatory, and antiproliferative effects [17,70,71]. Regrettably, these treatments are not curative and involve lifelong therapy with potential adverse effects [14,15,16,17,18].

##### NRTI Treatment

The six FDA-approved NRTI treatments of cHBV include Lamivudine (LMV, Epivir), Adefovir dipivoxil (ADV, Hepsera), Entecavir (ETV, Baraclude), Telbivudine (TBV, Tyzeka), Tenofovir disoproxil fumarate (TDF, Viread), and Tenofovir alafenamide fumarate (TAF, Vemlidy) (Figure 4A,B). NRTIs are prodrugs that need to be phosphorylated to their active 5′-triphosphate form by cellular kinases [14]. Once activated, NRTIs compete with natural dNTP substrates such as dATP, dGTP, dCTP, and dTTP for incorporation into the (−)-DNA strand by RT and function as DNA chain terminators.

NRTIs are administered at daily doses ranging from 0.5–600 mg. Clinical trial data from HBeAg-positive patients infected with cHBV indicate that after one year of NRTI treatment, 21–76% had an undetectable level of cccDNA, 41–77% had normalized levels of ALT, but only 12–22% achieved HBeAg seroconversion and 0–3% lost HBsAg [72,73,74,75,76,77]. Similar results were observed in HBeAg-negative patients [72,73,78,79]. Extending NRTI treatment to four to five years in both patient types led to an increase in HBeAg seroconversion (31–48%), but loss of HBsAg remained low (0–10%) [72,73,74,75,76,77,78,79]. Lifelong NRTI therapy could potentially reduce cHBV-related symptoms, but resistance to NRTIs limits their efficacy [14,15,16]. Resistance to NRTIs ranges from 0–80% after five years [72,73,75,76,78,79,80,81]. Moreover, NRTI treatment is associated with mild adverse effects, including headache, fatigue, and dizziness, as well as severe side effects such as increased liver toxicity, kidney tube dysfunction, myopathy, neuropathy, and decreased bone mineral density [82].

**Figure 4 viruses-15-01913-f004:**
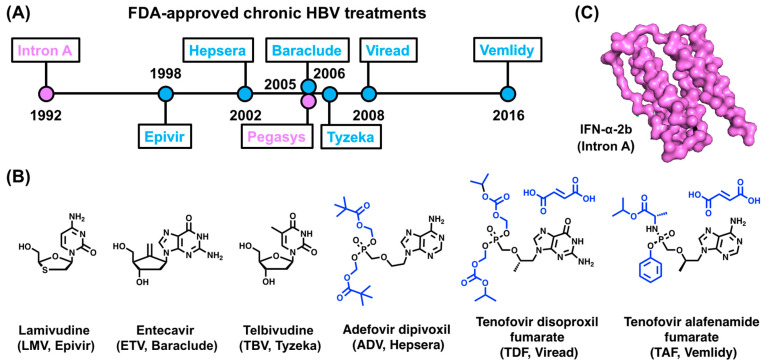
FDA-approved cHBV treatments. (**A**) Drug approval timeline. Trade names of NRTIs and IFN-α are shown in blue and purple, respectively. (**B**) Chemical structures of the NRTI prodrugs from (**A**), with regions that are removed during activation shown in blue. (**C**) Structure of IFN-α-2b (PDB 1rh2) [83]. Trade names for drugs in (**B**,**C**) are shown in parentheses.

##### IFN-α Treatment

The two FDA-approved IFN-α treatments for cHBV are IFN-α-2b (Intron A) and PEG-IFN-α-2a (Pegasys) (Figure 4A,C). IFN-α is a cytokine that is secreted by plasmacytoid dendritic cells [84] and is a soluble glycoprotein with potent antiviral activity [85,86,87]. IFN-α has been used to treat cHBV since 1976 and was the first FDA-approved cHBV treatment (Figure 4A). In 2005, PEGylated IFN-α-2a became the standard IFN-α treatment. Compared to NRTIs, less is known regarding the mode of action of IFN-α, which is thought to have dual roles. The first effect is to drive proliferation, activation, and antiviral potential of immune cells [17,70,71]. The second effect of IFN-α treatment is to inhibit (−)-DNA strand synthesis, which is supported by multiple lines of evidence. First, IFN-α treatment has been shown to increase the expression of the cytidine deaminase APOBEC3G [88], which inhibits (−)-DNA strand synthesis [89,90]. Second, IFN-α results in cccDNA-bound histone hypoacetylation and recruitment of transcriptional co-repressors to reduce cccDNA transcription [91].

PEG-IFN-α-2a treatment is administered through subcutaneous injections once a week [17]. Clinical trial data from HBeAg-positive patients infected with cHBV demonstrated that PEG-IFN-α-2a therapy was more effective than LMV, and the addition of LMV to PEG-IFN-α-2a treatment had no added benefit [92,93,94]. Specifically, a year of PEG-IFN-α-2a therapy led to modest levels of undetectable cccDNA (25%), normalization of ALT levels (34–39%), HBeAg seroconversion (27%), and negligible loss of HBsAg (3%) [92]. Similar results were observed in HBeAg-negative patients [95]. Extending PEG-IFN-α-2a therapy to four to five years resulted in varied outcomes, depending on HBV genotype and patient origin [96,97]. As with NRTIs, PEG-IFN-α-2a treatment also comes with mild side effects, including fatigue, flu-like symptoms, and mood changes, as well as severe adverse effects such as bone marrow suppression and autoimmune illnesses [17,18], and is susceptible to resistance [98]. Moreover, PEG-IFN-α-2a therapy has been reported to precipitate liver failure in patients with cirrhosis [18].

##### Perspective on Future NRTI and IFN-α Treatment

While NRTI and IFN-α therapies have been successfully employed to treat cHBV, both approaches have significant limitations. On the one hand, NRTIs are potent anti-HBV therapeutics with minimal adverse effects. However, NRTIs require lifelong treatment with resistance-related complications and cannot suppress genomic recycling of cccDNA [15,99,100]. IFN-α therapy can mitigate these burdens due to its finite treatment course and ability to modestly suppress cccDNA [101]. However, IFN-α has numerous adverse effects [17,18] and its subcutaneous administration is burdensome. Consequently, NRTIs with low resistance barriers and/or combinations of NRTI treatments aimed at reducing multidrug resistance holds the most promise for future prevention of HBV. 

#### 3.2.2. Alternative NRTIs in Clinical Trials for cHBV Treatment

In addition to the FDA-approved NRTIs (Figure 4A,B), there are other promising NRTIs that are approved or in clinical trials in the United States and other countries. A few examples of such NRTIs are Besifovir dipivoxyl maleate (BSV, Besivo), Tenofovir exalidex (TXL), and ATI-2173. BSV is a prodrug that was approved in South Korea in 2017. Clinical trial data showed that 48-week BSV therapy (150 mg) was as effective as ETV (0.5 mg) [102] and TDF (300 mg) [103] in treating cHBV. These findings were corroborated in a 96-week treatment with no observable drug resistance [103,104]. The only identified drawback of BSV was the depletion of L-carnitine in patients [104].

Another example of alternative NRTIs is TXL, which shares the Tenofovir (TEN) scaffold found in TDF and TAF but has been demonstrated to be approximately 100-fold more potent than TEN in vitro [105]. Initial clinical trials have administered TXL (up to 100 mg) to patients with cHBV and showed good tolerance and pharmacokinetics.

The final example is ATI-2173, an analog of Clevudine (CLV, Levovir). CLV was previously approved to treat cHBV in South Korea and the Philippines in 2006 but was later withdrawn due to complications with skeletal myopathy [106]. When active, CLV has a long half-life (11 h) and is a competitive RT inhibitor [107,108], suppressing HBV replication for months and reducing cccDNA levels in animal models [109]. Based on these findings, CLV analogs were reinvestigated. For example, ATI-2173 has shown in vitro efficacy against HBV, with significant reduction in cccDNA [110]. Consequently, a phase I clinical trial for ATI-2173 was recently initiated [111].

### 3.3. Alternative Anti-HBV Therapies

In addition to NRTI and IFN-α therapies, ongoing efforts are being made to develop additional anti-HBV therapeutics. Broadly speaking, these alternative therapies can be categorized by the macromolecule or viral event that is targeted. For simplicity, these therapeutic strategies will be listed in chronological order of the HBV replication cycle.

#### 3.3.1. Targeting Liver Cell Entry

The viral entry step is an attractive target for the development of new antiviral drugs [112,113,114]. Indeed, this strategy circumvents the shortcomings associated with NRTIs and IFN-α, as well as the difficulty in ridding the liver of cccDNA. The identification of NTCP as a bona fide HBV receptor [35] has attracted such therapeutic efforts as an ideal target. NTCP receptor function is inhibited by a variety of agents [112,113], including the well-characterized Myrcludex B, which is a synthetic *N*-acylated LHBs-derived lipoprotein [115,116,117,118]. Myrcludex B specifically targets hepatocytes and has been shown to efficiently block de novo HBV infection both in vitro [115,116] and in vivo [117]. Using human liver chimeric mice, it was demonstrated that Myrcludex B also prevents intrahepatic HBV spreading and cccDNA amplification [118]. Regrettably, this treatment strategy has some limitations. For example, current HBV NTCP inhibitors may interfere with native bile acid uptake, disrupting normal sodium-taurocholate transport in hepatocytes and inducing adverse effects [113]. NTCP-deficiency in patients has already been shown to cause hypercholanemia [119], and an increase in glycine-conjugated bile acid concentrations was even observed in patients treated with Myrcludex B [120]. Recent cryo-EM structures of the NTCP suggest that identifying small molecules that act as allosteric inhibitors of viral cell entry are of great therapeutic interest because they can stabilize the NTCP to block its function without eliciting an antigenic response as in Myrcludex B treatment [121,122].

#### 3.3.2. Targeting the Conversion of rcDNA to cccDNA 

Given that P protein release from the (−)-DNA strand of the rcDNA is required for its conversion into cccDNA, the inactivation of the responsible regulatory factor(s) should prevent cccDNA formation. One such factor is the DNA repair enzyme TDP2, which has recently been shown to cleave P from the rcDNA [43]. Moreover, using HepG2 cells, it was demonstrated that a reduction in TDP2 levels correlated with a slower rate of rcDNA to cccDNA conversion [43]. While these results are encouraging, two critical issues cast doubt on the efficacy of such treatments. The first important consideration is whether cccDNA longevity relates to individual molecules or is achieved via turnover [31]. In the absence of turnover, inhibiting rcDNA to cccDNA conversion would only work in the initial stage of infection [31].

The second critical issue of this treatment strategy is that targeting a DNA repair enzyme will likely trigger the native DNA damage response. On the one hand, if P release from the rcDNA can be achieved through multiple mechanisms, inhibiting TDP2 alone is unlikely to prevent cccDNA formation. However, this functional redundancy may help discover regulatory factors that are required for HBV but dispensable in the native DNA damage response. While this strategy requires extensive knowledge of the HFs involved, high-throughput screening may prove useful. As a first step in this direction, two small molecules that interfere with cccDNA formation were recently identified: the disubstituted sulfonamides CCC-0975 and CCC-0346 (Figure 5A) [123]. However, their exact mode of action remains unclear.

An alternative strategy that would circumvent the cccDNA-related issues would be to prevent the rcDNA from reaching the nucleus altogether. Given that nuclear transport depends on the nucleocapsid, this outcome can be achieved by capsid-targeting drugs, which are typically referred to as capsid assembly modulators (CAMs) and have been the topic of extensive review [34]. In addition to inducing the assembly of empty nucleocapsids or irregular polymers, these drugs may also destabilize existing mature nucleocapsids. Currently, there are 17 CAMs in recent and ongoing clinical trials and four more in current preclinical trials [34]. These CAMs encompass various chemotypes, including sulfamoylbenzamidine (SBA) (NVR 3–778) [124,125], sulfamoylpyrroloamides (SPA) (JNJ-56136379) [126,127], dibenzo-thiazepin-2-one (DBT) (ABI-H0731) [128,129], heteroaryldihydropyrimidine (HAP) (RO7049389) [130,131], amino-indane (AB-506) [132], and pyrazole (ZM-H1505R) [133] (Figure 5B). Similar to the limitations of NRTI and IFN-α therapies, CAM treatments are often plagued by limited potency, adverse effects, and resistance-related complications [34]. However, ongoing trials evaluating CAM treatment in combination with NRTI and/or PEG-IFN-α-2a support an enhanced antiviral efficacy compared to the respective individual therapies [124,134,135].

#### 3.3.3. Targeting the Transcriptional Activity of cccDNA

Rather than eliminating the cccDNA, an alternative therapeutic strategy is to leverage the cell’s epigenetic machinery to functionally inactivate the cccDNA. One validated method to achieve this end is with IFN-α treatment [91,136]. In one study using a chicken hepatoma cell line with an inducible duck HBV, IFN-α treatment led to a profound and long-lasting suppression of cccDNA by reducing acetylation of cccDNA-bound histone H3 lysine 9 and 27 [136]. In a second study, administration of IFN-α induced cccDNA-bound histone hypoacetylation and recruitment of transcriptional co-repressors to reduce cccDNA transcription in cells cultured with replication HBV and mice whose livers have been repopulated with HBV-infected human hepatocytes [91].

Interestingly, the HBV X protein was also found to be recruited to the cccDNA [137], suggesting that it may be an attractive therapeutic target in its own right. Indeed, mutations that prevent the expression of X led to hypoacetylation of cccDNA-bound histones and the recruitment of histone deacetylases, resulting in a significant reduction in transcribed pgRNA [137]. However, these X-mediated processes likely involve a complex network of interacting HFs [138], so more research is needed to understand its precise role in the epigenetic manipulation of the cccDNA.

#### 3.3.4. Targeting the ε–P Interaction

Since the ε–P interaction initiates protein-primed reverse transcription [19,24,25,26] and P–pgRNA packaging [7,27] (Figure 2A), the complex is traditionally pursued as an attractive therapeutic target for early intervention of HBV replication. Examples of such compounds include the antibiotic Geldanamycin (GDN), rosmarinic acid (ROS) derivatives, and iron protoporphyrin IX (Hemin, HEM), along with related porphyrin compounds (Figure 6A). In the first example, GDN disrupts the ε–P association in both human and duck models by inhibiting the function of the heat shock protein 90 (Hsp90) complex [139,140,141]. However, given the critical functions of Hsp90, this mode of preventing HBV replication is disadvantageous, prompting the search for additional compounds that target the ε–P complex. A second example involves ROS and its analog Quercetin (QUE) (Figure 6A), which specifically inhibit ε–P binding [142]. When combined with LAM, ROS slightly increased the anti-HBV activity of LAM, suggesting that ROS inhibition affects a replication step distinct from (−)-DNA strand elongation [142]. As a third example, HEM, Protoporphyrin IX (PPP-IX), Protoporphyrin IX disodium (PPP-IX-Na), and Biliverdin (BIL) (Figure 6A) all disrupt the ε–P complex in both duck and human models by binding to the TP domain of P [143]. Finally, (Z)-2-(allylamino)-4-amino-N′-cyanothiazole-5-carboximidamide (AACC) (Figure 6A) was shown to inhibit the ε–P interaction, which significantly reduced P–pgRNA packaging and blocked nucleocapsid assembly in multiple HBV genotypes [144]. Moreover, AACC inhibited the replication of LAM- and capsid-inhibitor-resistant HBV and showed synergistic effects with NRTIs and a capsid inhibitor [144].

Another approach to target the ε–P interaction is the Systematic Evolution of Ligands by EXponential Enrichment (SELEX) method to select strong P-binding RNAs that compete with ε for P binding [145]. These “decoy” RNAs show a strong inhibitory effect on P-pgRNA packaging and DNA synthesis [145]. While these ε–P inhibitors can complement existing NRTI treatment, the lack of structural data for the ε–P complex prevents the structure-informed design of anti-HBV therapeutics.

#### 3.3.5. Targeting Protein Priming

An alternative target of emerging anti-HBV therapies is protein priming, which offers the ability to prevent HBV replication at a very early stage. P is presumed to undergo a conformational change to transition from protein priming to the subsequent (−)-DNA strand elongation. This characteristic provides the opportunity to design anti-HBV therapeutic to specifically inhibit TP-mediated protein priming functions, complementing current NRTI treatments [9,146,147,148]. The FDA-approved guanosine analog ETV has been demonstrated to inhibit protein priming by competing with dGTP, the initiating substrate for synthesizing the 5′-GAA-3′ DNA (Figure 2A) [149,150]. Similarly, the adenosine analog TEN can also inhibit the elongation of the 5′-GAA-3′ DNA by competing with dATP [107]. Intriguingly, the thymidine analog CLV was also shown to inhibit protein priming through a noncompetitive mechanism without its incorporation into the (−)-DNA strand [107]. Using a related duck model, it has also been demonstrated that when added in trans, a catalytically inactive RT can inhibit protein priming, presumably by preventing the necessary interactions and/or conformations of TP and/or ε [151]. Whether this is feasible in human HBV remains to be seen. Nevertheless, it presents a novel targeting strategy.

#### 3.3.6. Targeting RNase H Activity

Most anti-HBV therapeutics target the RT domain of P. However, the RH domain of P has also emerged as a promising therapeutic target. RH degrades the pgRNA as RT elongates the (−)-DNA strand (Figure 2A). Consequently, blocking RH activity has been shown to prematurely halt (−)-DNA strand extension, resulting in the accumulation of extensive RNA–DNA hybrids that further inhibit (−)-DNA synthesis [152,153]. The recent development of active recombinant RH [152,154] has enabled low- and mid-throughput screening efforts to identify HBV replication inhibitors. These efforts have primarily focused on chemotypes that are known to inhibit human immunodeficiency virus (HIV-1) [152]. Screening of over 3000 compounds led to the identification of approximately 150 HBV replication inhibitors that function by blocking RH activity, as confirmed by the detection of RNA–DNA hybrid accumulation [152,155,156,157,158,159]. These inhibitors can be grouped into four chemotypes: *N*-Hydroxyisoquinolinediones (HID), *N*-Hydroxynapthyridinones (HNO), *N*-Hydroxypyridinediones (HPD), and α-Hydroxytropolones (αHT) (Figure 6B). Importantly, the efficacy of RH inhibitors against recombinant RH and clinical isolates from three HBV genotypes suggests that the genetic diversity of HBV is unlikely to complicate treatment [160].

### 3.4. HBV cccDNA: Key Obstacle for Curative HBV Treatments

One major factor of HBV persistence is a defective immune response, including the depletion of cytotoxic T cells, lack of CD4+ T cell help, and the failure to produce neutralizing antibodies [161,162,163,164]. However, the predominant virological basis to persistent cHBV infection, and a key obstacle for curative HBV treatments, is the cccDNA. While reliable kinetics of cccDNA loss are difficult to assess from human patients, the frequent rebound of HBV replication upon stopping NRTI therapy or immunosuppression suggest that cccDNA can persist for decades [165]. Studies on the related duck HBV system [44,45] have indicated cccDNA half-lives of ~30–60 days, and similar values are observed in primary hepatocytes [166]. Consequently, a cure for cHBV requires the elimination of the cccDNA. However, despite its critical role in HBV replication, a mechanistic understanding of cccDNA formation and degradation is lacking, owing to the lack of robust experimental systems [31]. Given the narrow host range of HBV, the most reliable infection system to study cccDNA formation is human primary hepatocytes, which are highly variable and not widely available [30]. Nevertheless, using these systems to deconvolute the steady-state cccDNA levels is paramount, as quantifying the lifetime of individual cccDNA molecules is crucial for the development of new therapeutics.

The most straightforward approach to rid the liver of cHBV infection is active elimination of existing cccDNA. One strategy to achieve this outcome is to mimic the immune-mediated clearance of cccDNA during acute HBV infection [164,167]. The two mechanisms of cccDNA clearance from hepatocytes are to “cure” or “kill”. The former option denotes the nondestructive elimination (i.e., “cure”) of cccDNA-containing cells, whereas the latter refers to their destruction by T cells (i.e., “kill”) and replacement by noninfected cells [167]. While both mechanisms likely exist, their respective contributions are not well understood due to the presence of additional parameters [168,169,170]. Nevertheless, cytokines such as IFs appear to play an important role in immune-mediated cccDNA clearance, though their exact mode of action is poorly understood. For example, in a study using HBV-infected cells, primary hepatocytes, and human liver needle biopsies, IFN-α treatment and lymphotoxin-β receptor activation upregulated APOBEC3CA and APOBEC3CB, resulting in cccDNA degradation that prevented HBV reactivation [87]. While these treatments are limited by systemic adverse effects [17,18], activating innate responses is a worthy goal. Promising preclinical results with the Toll-like receptor 7 agonist GS-9620 exemplify this approach [171].

An alternative approach to cccDNA elimination is genome editing with designer nucleases [172]. This strategy has been used with Zinc-finger nucleases [173,174], transcription activator-like nucleases [175], and CRISPR/Cas systems [176] to target cccDNA. However, many limitations remain, most notably the efficient access of the nucleases to edit all cccDNA molecules. In the absence of directly targeting cccDNA-containing cells, the nucleases must be delivered to all hepatocytes, opening the door to off-target effects that may compromise liver function [31]. In addition, it is unclear how targeting efficiency will be affected by excess rcDNA within the same cell [31]. Consequently, more work is needed to realize the potential and limitations of these genome editing strategies.

## 4. ε as an Underexploited and Attractive Therapeutic Target

Given that NRTI and IFN-α therapies are not curative and require lifelong treatment [14,15,16,17,18], the development of alternative anti-HBV therapeutics is imperative. Much of the work in this area has focused on P, but the lack of structural information has hindered detailed structure-informed drug design. However, the recent availability of structural data for ε [28] has motivated renewed interest in its potential as a therapeutic target. Indeed, the centrality of ε in HBV replication makes its inhibition an effective antiviral strategy. Consequently, a greater exploration of ε as a therapeutic target may help prevent cHBV infection and introduce a paradigm shift in current treatment strategies.

### 4.1. Early Characterization of ε

The secondary structure of a 61 nt ε has been determined (Figure 7A) [21,22], and its role in P binding [146,177], P–pgRNA packaging [21,22,25,178], and DNA synthesis [19,25,146,178] has been established by biochemical and mutational analyses. A comprehensive analysis of the functions and interactions of ε in each of these processes is provided in Section 2. This ε construct encompassing the entire stem-loop region is hereby referred to as full-length ε. Furthermore, the ε sequence is highly conserved among other mammalian Hepadnaviruses, between different isolates, and in more distantly related viruses [179,180]. While secondary structure analysis serves a useful starting point, a detailed understanding necessitates high-resolution structural information.

Fortunately, the solution NMR-derived structure of the truncated 27 nt ε upper helix (UH) [181,182] has revealed that its 5′-CUGUGC-3′ apical loop (AL) folds into a UGU triloop with a GC closing base pair and bulged out C, forming a pseudo-triloop (PTL) motif (Figure 7B). Additionally, the structure reveals that the entire UH forms a contiguous A-helix, with the exception of the PTL and U43 (using full-length numbering) bulge (Figure 7B) [182], suggesting that these noncanonical features may represent important anchor points for the initial recognition and binding of P.

**Figure 7 viruses-15-01913-f007:**
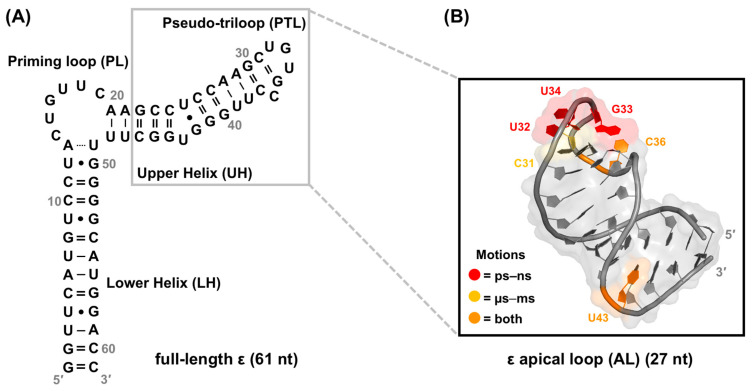
Summary of early NMR studies of ε. (**A**) Secondary structure of full-length ε [21,22] with abbreviations for structure elements that will be used throughout the text. The dotted line between A13–U49 signifies the absence of unambiguous evidence of base pairing from NMR measurements [28] (**B**) Top-ranked AL ε solution NMR conformer (PDB 2ixy) [182] with NMR data [183] mapped onto the structure. The AL comprises nucleotides in the UH and PTL, as indicated by the gray box in (**A**). Nucleotides in the PTL and U43 bulge (based on full-length numbering) exhibit both ps–ns and µs–ms motions, which have been hypothesized to facilitate P binding.

In conjunction with structural data, previous NMR studies have also assessed the dynamics of AL ε [183]. Specifically, picosecond (ps) to nanosecond (ns) motions were observed in PTL nucleotides U32–U34 and C36, as well as the U43 bulge, and microsecond (µs) to millisecond (ms) motions were detected in C31, C36, and U43 (all based on full-length numbering) (Figure 7B) [183]. These motions are hypothesized to facilitate P binding [183], presumably through conformational selection. Interestingly, SELEX experiments indicate that a variety of ε sequences are compatible with P binding, suggesting that structural flexibility within ε may help in recognizing and engaging P [145]. Taken together, these results imply that RNA dynamics play a crucial role in the ε–P interaction and, more broadly, in HBV replication. However, a comprehensive understanding of this mechanism requires further insight into P protein structure, which is currently lacking.

### 4.2. P Protein Structure and Host Interactions

Currently, there are no structures of HBV or Hepadnaviral P proteins, although homology models have been proposed for the RT [184] and RH [185] domain. The RT model agrees with drug resistance data and is corroborated by mutational analyses [186]. However, beyond the active site, the accuracy of these models remains uncertain. Regrettably, the lack of structural information significantly limits our mechanistic understanding of how P interacts with HFs and ε, hindering the rational design of effective therapeutics. The situation for the TP domain is even more challenging, as it does not share significant sequence similarity to other proteins, including the few other TPs involved in viral genome replication [56,187]. Moreover, these TPs are not covalently linked to their polymerases, rendering homology-based efforts of limited utility. Consequently, structure determination of P or its individual domains is desirable, though it has proven to be a formidable task. While the inclusion of solubility fusion partners has partly mitigated this challenge, they form soluble aggregates at high concentrations [187,188,189]. To compensate for this dearth of structural data, ab initio predicted models have recently been reported for the TP domain [190] and the entire P protein [191]. These efforts, while promising, can only serve as provisional solutions until more accurate structural data is obtained, reinforcing the need for continued research in this area.

Despite the limited structural understanding of P, an extensive body of knowledge exists about HFs that modulate HBV and Hepadnaviral P protein function. The first HF demonstrated to interact with P is the Hsp90 complex, which includes Hsp90, Hsp70, Hsp40, Hop/p60, and p23 [139,140,141]. This complex is essential for establishing and maintaining the P conformation that binds ε [139,140,141]. In addition, and akin to (−)-DNA strand synthesis, eIF4E (not to be confused with the eIF4G mentioned in Section 2.3) binds to P [192]. This interaction can occur in an RNA-independent manner, though the presence of the pgRNA enhances P–eIF4E binding [192]. Another RNA-independent binding partner is APOBEC3G [193], which inhibits the early stages of (−)-DNA strand synthesis [89,90]. HBV also forms an intricate interaction network with the host immune response. For example, the immune modulatory DEAD-box RNA helicase 3 (DDX3) has been shown to interact with P in an RNA-independent manner [194,195], but its precise function remains unclear. Additional studies have revealed that P binds to nuclear translocation proteins importin-α5 and protein kinase C-δ (PKC-δ) [196]. Further research is necessary to elucidate how these proteins may regulate HBV replication.

### 4.3. Essential Factors and Dynamic Underpinnings of the ε–P Interaction for HBV Replication

The fundamental prerequisites for the ε–P interaction [146,177], P–pgRNA packaging [21,22,25,178], and DNA synthesis [19,25,146,178] have been elucidated through biochemical and mutational analyses. With regards to proteins, the TP and RT domains of P are necessary for ε–P binding [146,177,197] (Figure 8A). Cellular chaperones such as the Hsp90 complex [139,140,141] and HFs, including eIF4E [192], APOBEC3G [89,90,193], DDX3 [194,195], importin-α5 [196], and PKC-δ [196], play crucial roles in facilitating the ε interaction and subsequent P–pgRNA packaging.

In the context of the pgRNA, an essential proximity between the 5′-end ε and the 5′-cap is required for efficient reverse transcription [50]. Moreover, mutational studies [19,21,22,25,146,177,178] have demonstrated that various ε regions contribute to P binding, protein priming, and P–pgRNA packaging in both a sequence- and structure-dependent manner (Figure 8B). Specifically, the upper segment of the lower helix (LH) exhibits primary sequence requirements for P binding [177] and P–pgRNA packaging [21,25]. Intriguingly, the lower segment of the UH has sequence requirements for P binding [177] and DNA synthesis on its 5′-side [25], and P–pgRNA packaging on its 3′-side [21,25], while the upper segment of the UH primarily serves a structural role [21]. The PL bulge structure is indispensable for P binding [177], P–pgRNA packaging [19,21,22,146,177], and DNA synthesis [21,22,25,146], with its 5′- and 3′-ends functioning distinctly in P binding [177] and protein priming [25], respectively. Additionally, the PTL [21,177] and U43 bulge [22,177] are essential for P–pgRNA packaging, though only the PTL is dispensable for P binding [177]. Lastly, the AU base pairs at the base of the UH are required for protein priming and (−)-DNA strand elongation, respectively [178].

While these findings offer valuable insights, the biochemical methods are opaque to quantifying dynamic interactions and processes. Moreover, the mutational work may lead to unintended perturbations outside the regions being probed. Consequently, achieving a comprehensive mechanistic understanding of the ε–P interaction and its subsequent functions necessitates building on these findings with high-resolution structural dynamics studies. As previously mentioned, such data are entirely absent for P but have recently become available for full-length ε [28,198].

### 4.4. Structural Dynamics Characterization of Full-Length ε

Our group recently determined the structure of full-length ε using a combination of small-angle X-ray scattering (SAXS) and solution NMR spectroscopy (Figure 9A) [28]. Our SAXS- and NMR-derived model aligns well with the previously determined AL ε NMR structure (Figure 9B) [28,181,182] and offers perspectives on how the 6 nt PL bulge might recognize and engage P. Particularly, in three of the top-ten-ranked NMR structures, nucleotides U15–C19 remain well oriented, with G16 and U17 partially stacked with an evident backbone kink centered between nucleotides C14–G16 (Figure 9A).

Interestingly, in vitro biochemical experiments suggest that nucleotides C14–G16 likely have a role in stalling the HBV replication complex, which potentially acts as a prerequisite for (−)-DNA strand transfer [19]. Although no structural data exist for P, the consistent topology of all DNA and RNA polymerases suggest a universal architecture, assembling into a structure similar to a right hand with “fingers”, “palm”, and “thumb” domains [181,182]. The palm domain catalyzes phosphoryl transfer, the fingers interact with the incoming dNTP and the template nucleotide to which the dNTP is base-paired, and the thumb domain assists in positioning of the DNA–RNA hybrid, processivity, and translocation, and in potentially acting as a sensor of nucleic acid conformation [181,182]. Consequently, it is plausible that the stacking and unstacking of G16 and U17 could facilitate initiation of (−)-DNA strand elongation, while the thumb’s detection of the kink turn at U15 could arrest translocation, triggering the first strand transfer event. Although this model remains conjectural, our structural data suggest this possibility. 

Alongside structural measurements, our preliminary NMR dynamics data suggest that nucleotides in the PL (C14–C19 and adjacent U49), PTL (U32–U34 and C36), and U43 bulge undergo ps–ns motions (Figure 9C) [28], aligning with previous studies on the AL ε (Figure 7B) [183]. However, the extent to which full-length ε undergoes motion on additional timescales or adopts multiple conformations, and how these dynamics influence the ε–P binding interaction, remains unclear.

To begin evaluating the relevance of RNA dynamics for the ε–P interaction and its downstream functions, we combined solution NMR with molecular dynamics (MD) simulations to probe motions on the ps–ns, ns–1 μs, and μs–ms timescales. In agreement with our recent work (Figure 9C) [28] and previous NMR studies (Figure 7B) [183], nucleotides in the PL (C14–C19 and adjacent U48 and U49), PTL (U32–C36 and adjacent G30), and U43 bulge (and adjacent G41 and G42) undergo ps–ns motions (Figure 9E) [198]. In addition, nucleotides in the PL (C14, U15, U17–C19, U48, and U49), PTL (U32, U34, and C36), and U43 bulge all experience μs–ms conformational exchange that is fast on the NMR timescale (Figure 9D), albeit with diverse exchange rates and lifetimes [198]. These findings suggest that slower motions are overlaid onto faster ones, with the noteworthy exception of G16 and G33 in the PL and PTL, respectively (Figure 9D) [198]. MD data further reveal that ε undergoes vast conformational dynamics beyond the timescale of sampling (i.e., 1 μs), which is mainly driven by the flexibility of the PL [198].

Together, these data outline a series of complex motions on multiple timescales within the PL, PTL, and U43 bulge (Figure 9D). Moreover, these motions are localized to nucleotides that are highly conserved (Figure 9E) [179,180] and functionally important (Figure 8B). For example, every dynamic nucleotide in the PL, PTL, and U43 bulge is 97% conserved except for G16 and G42 and C31, which are approximately 75% and 90% conserved, respectively (Figure 9D,E). While nucleotides in rigid structural elements (e.g., helices) may be conserved to maintain their structure, the motions in the nonhelical PL, PTL, and U43 bulge nucleotides are likely conserved for their biological function.

### 4.5. ε Dynamics as a Regulatory Component of the ε–P Interaction in HBV Replication

The diversity of RNA sequence- and structure-specific requirements for HBV replication (Figure 8B) suggests that each step (i.e., P binding, protein priming, P–pgRNA packaging, and DNA synthesis) may necessitate a distinct conformation of ε and P. Moreover, ε assumes conformational states beyond unbound, P-bound, and priming-competent, as demonstrated by the case where ε–P binding occurs without protein priming [177]. Consequently, ε dynamics may enable binding to P and cellular HFs, and facilitate conformational changes required for protein priming, P–pgRNA packaging, and DNA synthesis (Figure 10), as suggested in the related duck HBV model [29,48,203]. 

Within this context, we posit that dynamics may correlate to function as follows. Considering that the PL and U43 bulge are required for P binding (Figure 8B) [146,177], nucleotides exhibiting fast ps–ns motions (Figure 9C,D) in these regions could facilitate this interaction (Figure 10). Though the PTL is not essential for P binding (Figure 8B) [177], it is proposed to interact with cellular HFs, which are necessary for protein priming and P–pgRNA packaging [89,90,139,140,141,192,193,194,196]. Similarly, ps–ns motions in the PTL (Figure 9C,D) may promote binding to HFs (Figure 10). Both scenarios involve conformational selection mechanisms, where mobile nucleotides facilitate binding interactions. Superimposed on these fast motions, it is plausible that a series of intricate conformational changes could modify contact networks between ε, P, and HFs to encourage protein priming, P–pgRNA packaging, and DNA synthesis (Figure 10). Fast μs–ms exchange processes in PTL nucleotides might initiate the transition of the ε–P–HF complex to its priming-competent state (Figure 10). Subsequent slower exchange events may position ε within the P–HF complex to present the 5′-end of the PL to the TP domain for priming and initiating reverse transcription (Figure 10). While this model is speculative, it is consistent with our data. 

In conclusion, our combined NMR and MD data reveal a series of complex motions on multiple timescales within full-length ε (Figure 9C,D). This work serves as a valuable extension to our recent ε structure elucidation (Figure 9A) [28] and highlights a critical theme: ε does not adopt a single, stable structure, but rather exhibits a dynamic conformational ensemble. The motions occur in conserved structural regions (Figure 9E) that are functionally important (Figure 8B). Collectively, our findings suggest that ε dynamics may be an integral component of HBV replication. This proposed dynamic model (Figure 10) would benefit from NMR measurements of ε in the presence of P or, at minimum, its RT and TP domains, which is currently not feasible, although recent advances in cryo-EM [60,61,62] may soon change this. Consequently, our dynamic characterization of ε serves as a fundamental starting point for a more detailed understanding of how RNA dynamics regulate HBV replication.

## 5. Discovery of ε-Targeting Small Molecules 

Small molecules present an opportunity to target RNA motifs such as pseudoknots, bulges, and hairpins, which are often highly conserved and mediate important biological functions [204,205,206,207]. Several recent in vitro [208] and in silico [209] high-throughput screening (HTS) methods have identified chemotypes that selectively bind RNA motifs with physiological effects in cell culture and animal models. Given the significance of the ε–P interaction in HBV replication [19,21,22,25,146,177,178,197], ε is an attractive therapeutic target. Our structural analysis of full-length ε suggests that the 6 nt PL bulge forms a binding pocket that is amenable to small molecule targeting (Figure 9A).

### 5.1. HTS Strategy

As an initial step in testing our structure-informed hypothesis, we employed a small molecule microarray (SMM) approach, which has been previously employed to identify various chemotypes targeting RNA [210,211,212,213] and DNA [214,215] motifs [28]. Here, fluorescently tagged full-length ε and a control RNA were used to screen a library of approximately 26,000 compounds. With the SMM-identified ε-binding compounds, five potential leads were selected based on pharmacophore properties (i.e., tunability via medicinal chemistry) and commercial availability (Figure 11A).

Subsequently, NMR titration experiments were employed to distinguish specific binders from nonbinders, aggregators, and nonspecific binders. This analysis revealed that Raloxifene selectively targets the ε PL [28]. Raloxifene, a benzothiophene (Figure 11A), belongs to the class of selective estrogen receptor modulators (SERMs) and is clinically used for the treatment of osteoporosis by mimicking the effects of the hormone estrogen to increase bone density. Raloxifene is also suggested to lower the risk of breast cancer by blocking the effects of estrogen on breast tissue [216]. The benzothiophene, Arzoxifene, and the phenylindole, Bazedoxifene, are closely related SERMs that are also under clinical investigation (Figure 11B) [217]. To test whether these compounds bind to full-length ε, NMR titrations were performed. This analysis revealed that Bazedoxifene also targets the ε PL, whereas Arzoxifene does not bind (Figure 11B) [28].

As an independent measure of SERM binding, we used a dye-displacement assay. These data corroborate the NMR titration data and suggest that Raloxifene and Bazedoxifene have affinities (measured by IC_50_) of approximately 70 μM and 110 μM, respectively, whereas Arzoxifene does not bind (Figure 11B) [28]. Although obtaining an NMR structure of Raloxifene-bound ε is desirable, saturation of the RNA was not feasible at the required NMR concentrations due to Raloxifene’s insolubility. 

To gain further insight into the ε–Raloxifene interaction, computational docking and MD simulations were conducted. Raloxifene docking pose predictions indicated that among the top-10 NMR ε conformers [28], three could be targeted directly at the PL, with ε R3 scoring best, ε R6 a close second, and ε R5 third. Notably, these are the three ε conformers that share the unique PL orientation (Figure 8A). The predicted docking pose reveals that Raloxifene’s core is deeply wedged into the PL between nucleotides U15–C19 (with G16, U18, and C19 rotated away) and is in close proximity to U48 and U49 (Figure 11C). Interestingly, MD simulations reveal that Raloxifene binding modulates ε motions, as it rigidifies the conformational dynamics of the ε PL.

A detailed view of the docking pose of the ε–Raloxifene complex reveals that the hydroxyethylpiperidine tail occupies the groove of ε and seldom interacts with the PL binding pocket (Figure 11C). This led us to hypothesize that removal of the tail and/or other chemical modifications could enhance small molecule binding to ε. To this end, several Raloxifene analogs were synthesized and their affinities were determined by our dye-displacement assay. We divided Raloxifene into three “units”: (i) a hydroxyethylpiperidine tail, (ii) a 3-(carbonyl) position hinge, and (iii) a 6-,4′-substituted phenylbenzothiophene (Figure 11A). Regarding (iii), replacing the 4′-OH with -Br (SG70) or -OCH_3_ (SG74) severely reduced affinity (Figure 11D). In the presence of (i), replacing the (ii) with -OH (SG102) or removing it (SG113) decreased affinity (Figure 11D) [28]. Interestingly, removing (i) (SG92) increased affinity approximately two-fold (Figure 11D), suggesting that the tail is dispensable, in agreement with our docking observations [28].

Based on these results [28], Raloxifene and SG92 were used in a cell culture assay to evaluate whether they could prevent HBV protein priming [218]. In this experiment, ε and P were transfected into cells and treated with Raloxifene and radiolabeled [α-^32^P]-dGTP. Given that dGTP initiates the synthesis of the 5′-GAA-3′ DNA (Figure 2A), successful protein priming can be detected by the incorporation of [α-^32^P]-GTP into the (−)-DNA strand by phosphorimaging. Unfortunately, neither Raloxifene nor SG92 affected HBV protein priming [218], motivating the search for additional ε-targeting ligands.

### 5.2. Virtual Screen Strategy

As a second method to test our structure-informed hypothesis, we employed a structure-based virtual screening (VS) approach [219]. Computational docking can provide complementary data and corroborating evidence to experimental binding assays. Moreover, VS greatly reduces the amount of time to generate lead compounds. Given the inherently flexibility of RNA, it is advisable to treat the RNA target as a conformational ensemble that is then subject to VS [220,221,222]. These ensembles can be computationally derived or experimentally informed [209,223,224,225,226,227,228]. The latter approach has shown initial success in an ensemble-based VS, suggesting a promising direction for RNA [209,223]. However, the efficacy of this method depends on robust and extensive experimental constraints, which are lacking for full-length ε due to its large size.

As an alternative, we employed a rigid dock VS followed by MD simulations as a means to rapidly identify lead compounds while partially accounting for the inherent dynamics of ε [28,198]. Based on our prior computational docking [28], we utilized our full-length ε R3 structure (PDB 6var) [28] as the receptor. We then selected a 1604-compound FDA-approved library curated on the ZINC15 database [229,230,231]. With our receptor and library set, we executed VS to identify FDA-approved drugs that could be repurposed as anti-HBV therapeutics [219]. We implemented selection criteria based on affinity, commercial availability, drug-like properties, and docking site [219]. Ultimately, we selected 12 potential lead compounds (Figure 12A) with diverse chemotypes and uses that could now be experimentally verified [219].

Initial lead compounds were then experimentally validated with a dye-displacement assay. Of the 12 compounds, nine showed no evidence of binding, whereas three compounds, Ledipasvir, Simeprevir, and Daclatasvir, did bind full-length ε with an affinity (measured by EC_50_) range of 60–300 µM (Figure 12A) [219]. We repeated our dye-displacement assay with additional RNA targets to determine whether these compounds selectively target ε or are merely nonspecific binders. This analysis revealed that only Daclatasvir selectively targeted ε. As an initial effort to map the specific binding site of Daclatasvir to full-length ε, we used our dye-displacement assay with three ε modular constructs, coupled with NMR titrations. Collectively, these data suggest that Daclatasvir binds full-length ε at its PL and upper segment of the LH (Figure 12B).

Similar to Raloxifene, compound insolubility precluded the use of NMR to acquire a Daclatasvir-bound full-length ε structure, so we once again employed computational docking and MD simulations. Our docking analysis revealed that Daclatasvir selectively targets the ε PL, with its core wedged between nucleotides U15 and U17–C19, and also contacting the adjacent A20–G22 and U47–G51 and C5 and A6 in the LH (Figure 12C) [219]. Interestingly, akin to Raloxifene, MD simulations reveal that Daclatasvir binding modulates ε dynamics, albeit in a different manner. Daclatasvir binding rigidifies the dynamics of nucleotides U17–C19 but increases the conformational variety of U15 [219].

### 5.3. RNA Dynamics as a Target for Future Anti-HBV Therapeutics

Interestingly, both Raloxifene and Daclatasvir have been shown to regulate the dynamics of the ε PL. As discussed in Section 4.4 and Section 4.5 [28,198], the structural dynamics of the PL nucleotides (Figure 9C,D) are essential to their role in facilitating HBV replication (Figure 10) [19,21,22,25,146,177]. Consequently, modulating ε dynamics might be an effective therapeutic strategy. This approach could benefit mid-µM binders that cannot directly outcompete P binding, as the latter exhibits significantly higher affinity (i.e., low nM) for ε. For example, small molecules that regulate dynamics could exert their effect by preventing ε from adopting the conformations necessary for transitioning between functional states (i.e., P binding, P–pgRNA packaging, and reverse transcription) (Figure 13). Indeed, taking RNA dynamics into account when targeting with small molecules has shown promising results in RNA-targeted drug discovery [209]. Therefore, even though Raloxifene was shown to have no anti-HBV effect, and this information is not yet available for Daclatasvir, the approaches outlined in Section 5.1 and Section 5.2 offer valuable platforms for the discovery of novel ε-targeting ligands. The ability of these compounds to alter ε dynamics could lead to the inhibition of the early stages of HBV replication.

## 6. Conclusions and Outlook

Approximately two billion people worldwide have been exposed to HBV. This global exposure has resulted in over 300 million cHBV infections and around 800,000 deaths annually [63,64,65], with the majority occurring in developing countries (Figure 3). Moreover, HBV is responsible for about 30% and 50% of all cases of cirrhosis and HCC, respectively [66]. This global burden has motivated intense efforts to discover and develop cHBV treatments. Currently, there are eight FDA-approved therapies (Figure 4A): two IFN-α treatments (Figure 4C) and six NRTIs (Figure 4B), with additional therapies in clinical trials. However, both treatments have limitations. IFN-α is associated with adverse effects and NRTIs require lifelong therapy with the risk of developing resistance [14,15,16,17,18]. Nevertheless, NRTIs remain the most effective drugs in combating cHBV infection. Still, there is a pressing need for additional anti-HBV therapeutics to complement existing NRTI treatments. Alternative targets include the NTCP receptor [115,116,117,118], cccDNA [90,119,120,121,122,123,124,125,126,127,128,129,132,133,134], ε–P interaction [142,143,144,145], protein priming [107,149,150], and RH domain of P [152,155,156,157,158,159,160].

However, the lack of structural information for the ε–P complex and P itself prevents structure-informed design of anti-HBV therapeutics. Our recent structure of full-length ε [28] (Figure 9A) is a crucial step in this direction. Considering the central role of ε in HBV replication (Figure 8B), it emerges as an attractive and novel therapeutic target. Our structural analysis of full-length ε suggests that the 6 nt PL bulge forms a binding pocket that is amenable to small molecule targeting (Figure 9A). To validate this hypothesis, we employed two strategies to identify ε-targeting ligands.

Our first approach involved an HTS-based approach (Section 5.1). We screened approximately 26,000 compounds using an SMM followed by NMR titrations to identify the SERM, Raloxifene, as an ε-targeting ligand with mid-µM affinity (Figure 11) [28]. However, cell culture assays showed that Raloxifene did not prevent protein priming [218]. As an alternative, we employed a VS method (Section 5.2), which circumvented the months to years required to generate lead compounds using HTS-based strategies. Through the computational screening of a 1604-compound FDA-approved drug library from the ZINC15 database [229,230,231], followed by subsequent binding experiments, we identified the anti-HCV drug Daclatasvir as another ε-targeting ligand with mid-µM affinity (Figure 12) [219]. As of now, the cell culture or in vivo effects of Daclatasvir on HBV remain unknown. Nevertheless, both ε-binding compounds were found to modulate ε dynamics, offering a novel route for anti-HBV therapeutic intervention (Figure 13).

Future experimental work in this area must focus on determining high-resolution structures of ε–small molecule complexes and employing structure–activity relationship studies to reveal chemotypes that selectively target the ε PL. Given the inherent dynamics of full-length ε [28,198], complementary computational work should focus on VS against an ε conformational ensemble. Indeed, it is essential for these structural and computational efforts to probe beyond static structures to investigate alternative conformations that are sampled on different timescales. This is particularly critical when considering the role of lowly populated states that, while transient, often play pivotal roles in molecular function. Consequently, such transient states have proven to be viable targets for effective therapeutic intervention [209,223]. Given our models of dynamic regulation (Figure 10 and Figure 13), we posit that targeting the ε conformation that is required for P binding is paramount. Together, these efforts can help discover and optimize dynamic modulating, ε-targeting small molecules. While this approach will likely face the difficulties associated with cccDNA persistence [31], we hope that, in combination with other therapeutic strategies, it can be helpful in achieving the ambitious goal of finding a cure for cHBV infection. 

## Figures and Tables

**Figure 2 viruses-15-01913-f002:**
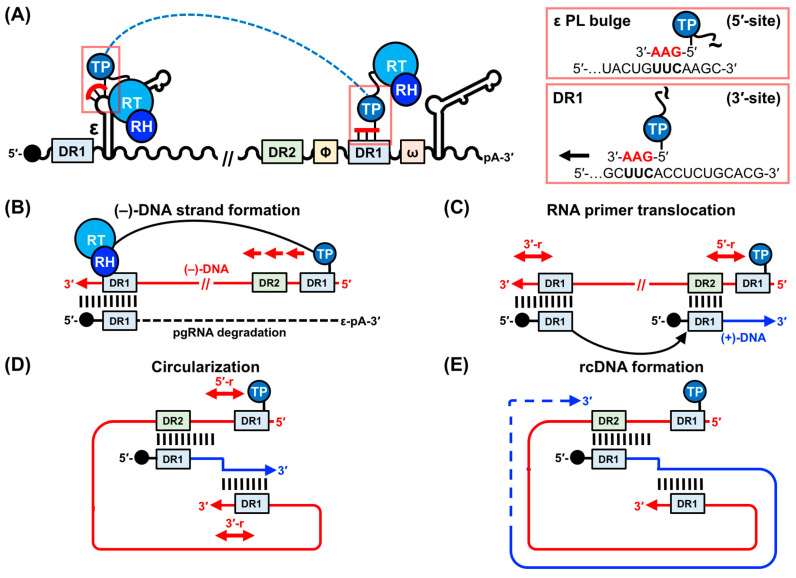
HBV genome replication. (**A**) The binding of ε to P initiates P–pgRNA packaging and subsequent reverse transcription. Before (−)-DNA strand formation, the TP domain synthesizes a 3 nt DNA (5′-GAA-3′), which is templated by the ε PL bulge (5′-UUC-3′). The TP-linked DNA then translocates (indicated by a dashed blue line) to the 3′-end DR1 motif (5′-UUC-3′) where (−)-DNA strand elongation begins. (**B**) The TP-linked DNA extends (−)-DNA strand synthesis toward the 5′-end of the pgRNA, which is concurrently degraded by the RH domain of P. (**C**) The RNA primer subsequently translocates to the DR2 motif and extends toward the 5′-end of the (−)-DNA strand, initiating (+)-DNA strand synthesis. Here, the terms 3′- and 5′-r refer to the 10 nt redundancy that is generated with the (−)-DNA strand. (**D**) After copying the 5′-r, the growing 3′-end of the (+)-DNA strand translocates to the 5′-r on the (−)-DNA strand to permit further elongation. (**E**) The final extension of the (−)-DNA strand template yields (+)-DNA strands of various lengths to form the new rcDNA. This figure is adapted from [29].

**Figure 3 viruses-15-01913-f003:**
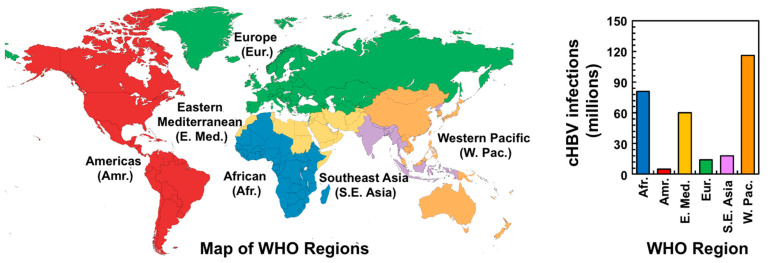
cHBV infection statistics. Map of the WHO regions (**left**) and their cHBV infection statistics (**right**). Data were accessed from the WHO website (https://www.who.int/news-room/fact-sheets/detail/hepatitis-b (accessed on 31 August 2023)).

**Figure 5 viruses-15-01913-f005:**
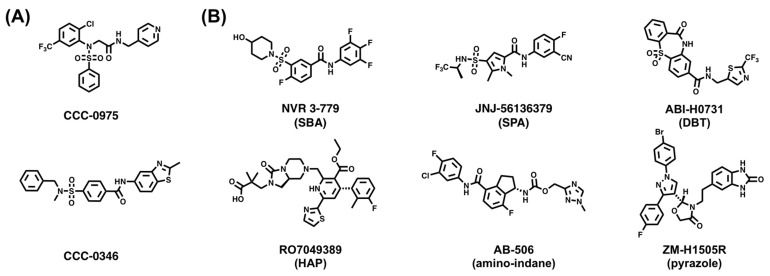
Small molecules targeting the conversion of HBV rcDNA to cccDNA. Chemical structures of (**A**) disubstituted sulfonamide cccDNA formation inhibitors and (**B**) CAMs in recent and ongoing clinical trials from the SBA, SPA, DBT, HAP, amino-indane, and pyrazole chemotypes.

**Figure 6 viruses-15-01913-f006:**
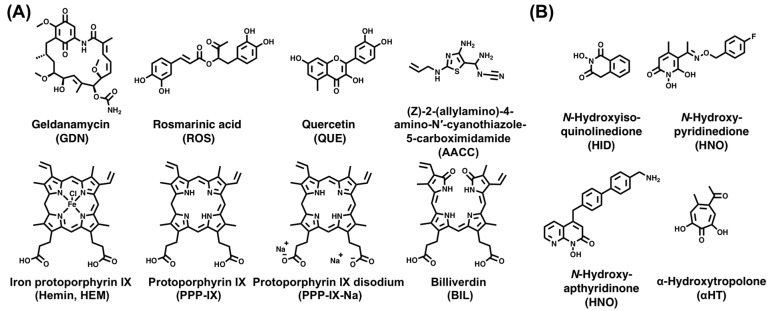
HBV P and ε–P complex targeting small molecules. Chemical structures of (**A**) ε–P complex inhibitors and (**B**) RH inhibitors from the HID, HNO, HPD, and αHT chemotypes.

**Figure 8 viruses-15-01913-f008:**
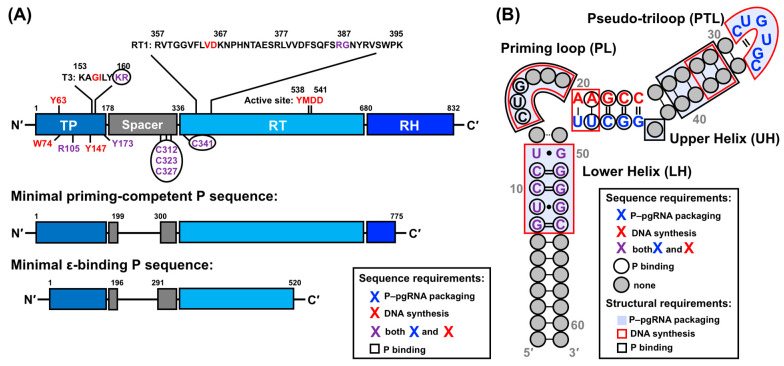
Sequence and structural dependencies of P protein and ε RNA in HBV replication. (**A**) The sequence and/or structural requirements of P for ε–P binding, specifically for the RT domain. (**B**) The sequence and/or secondary structure prerequisites of ε for ε–P binding, P–pgRNA packaging, and DNA synthesis. Depictions in (**A**,**B**) are based on a synthesis of prior biochemical and mutational studies [19,21,22,25,146,177,178,197].

**Figure 9 viruses-15-01913-f009:**
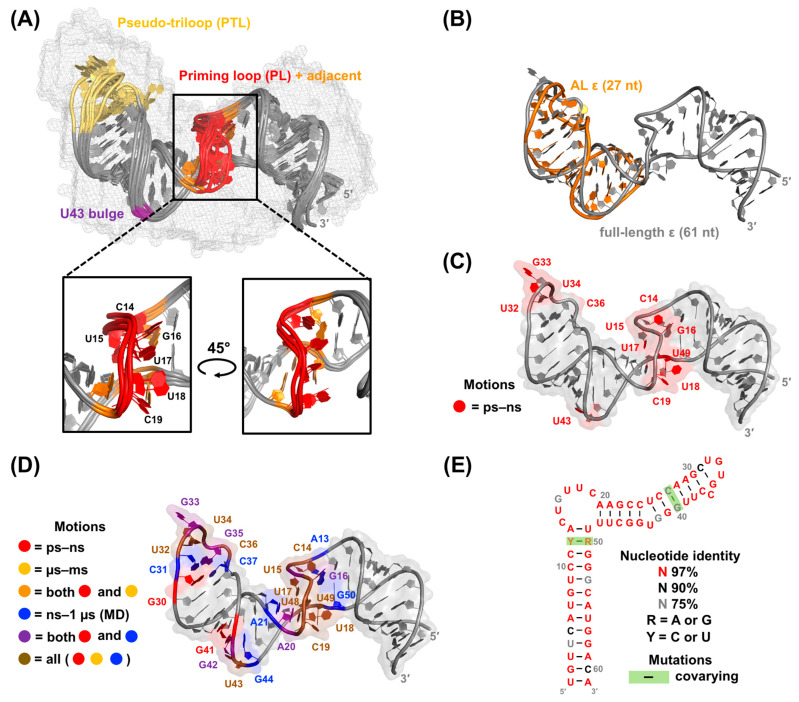
Summary of the structural dynamics data of full-length ε. (**A**) Bundle of the top-10 lowest energy ε structures (PDB 6var) [28] generated by Xplor-NIH [199], with an RMSD of 1.8 Å and displayed within the SAXS envelope and with important structural regions colored. A close-up view of the PL in three NMR conformers (rank 3, 5, and 6) is also shown. These conformers share the backbone kink centered at U15, followed by partially stacked G16 and U17. (**B**) Overlay of the top-ranked full-length ε NMR conformer (PDB 6var) [28] and AL ε NMR conformer (PDB 2ixy) [182]. Structures show strong agreement (RMSD of 1.7 Å). (**C**) Top-ranked full-length ε solution NMR conformer (PDB 6var) [28] with NMR dynamics data [28] mapped onto the structure. (**D**) Similar to (**C**) but from our recent combinined NMR and MD data [198]. (**E**) Rfam (RF01047) representation [200,201,202] of ε, showing structure and sequence conservation. Nucleotide identity calculations and covarying mutations are based on 36 sequences from six species, including different HBV genotypes but excluding the related avian HBV sequences (Rfam RF01313). As depicted in (**C**–**E**), highly conserved nucleotides in and adjacent to the PL, PTL, and U43 bulge exhibit motions across multiple timescales.

**Figure 10 viruses-15-01913-f010:**
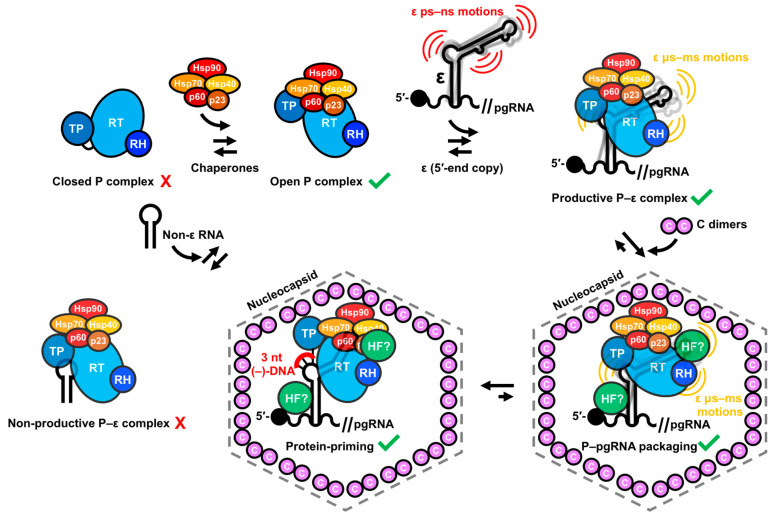
Schematic of the molecular and biophysical determinants of HBV replication. This model draws on mutational and biochemical data of P and ε [19,21,22,25,146,177,178,197] and our recent NMR structural dynamic studies of ε [28,198]. Additional details can be found in the text.

**Figure 11 viruses-15-01913-f011:**
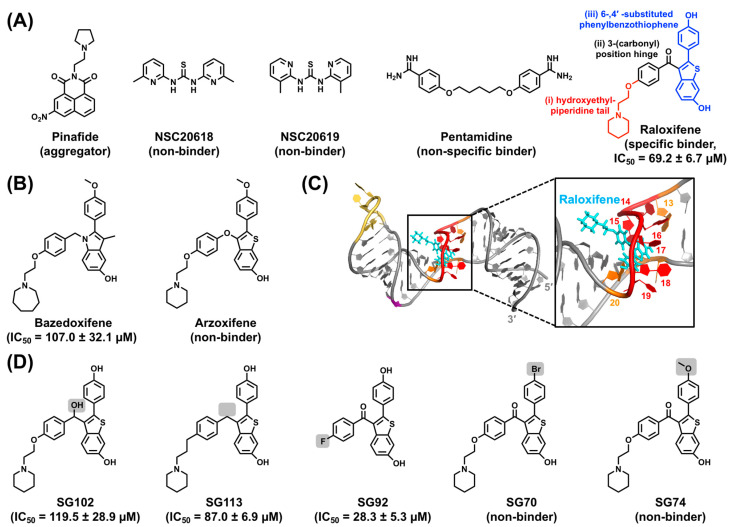
HTS of full-length ε. (**A**) Chemical structure of SMM-derived lead compounds with their NMR-derived binding mode (i.e., specific or nonspecific binder, nonbinder, or aggregator) and dye-displacement-derived affinity to full-length ε (measured by IC_50_) shown in parentheses. (**B**) Chemical structure of SERMs with their affinity to full-length ε reported as in (**A**). (**C**) Top-ranked Raloxifene docking pose to ε R3 (PDB 6var) [28], which is colored as in Figure 9A. Raloxifene is depicted in cyan sticks and interacting nucleotides are labeled. As seen from the close-up view of the binding pocket, the Raloxifene hydroxyethylpiperidine tail does not participate in binding. (**D**) Chemical structure of Raloxifene analog library with their affinity to full-length ε reported as in (**A**). Any modification to the three Raloxifene “units” (shown in (**A**)) is highlighted with a gray box.

**Figure 12 viruses-15-01913-f012:**
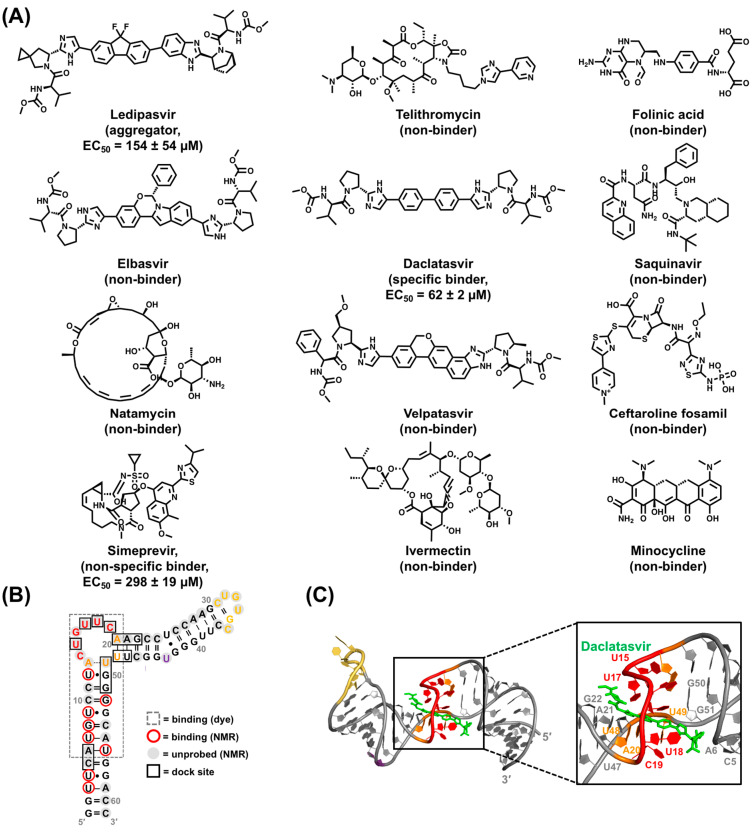
VS of full-length ε. (**A**) Chemical structures of VS-identified lead compounds with their dye-displacement derived binding mode (i.e., specific or nonspecific binder, nonbinder, or aggregator) and affinities to full-length ε (measured by EC_50_) shown in parentheses. (**B**). Representative data from dye-displacement, NMR titration, and computational docking analysis are shown mapped onto the secondary structure of ε. Collectively, these data agree that Daclatasvir targets FL ε mainly at its PL and upper segment of the LH. (**C**) Top-ranked Daclatasvir docking pose to ε R3 (PDB 6var) [28]. Daclatasvir is depicted in green sticks and interacting nucleotides are labeled. Structure representations in (**B**,**C**) are colored as in Figure 9A.

**Figure 13 viruses-15-01913-f013:**
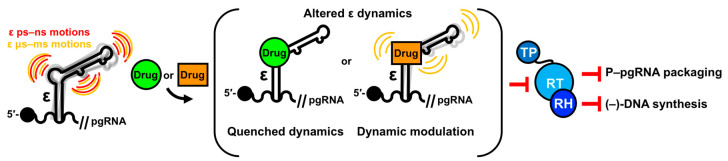
Schematic of dynamic-regulating, ε-targeting small molecules. This model draws on our recent NMR structural dynamic studies ε and ligand-bound computational modeling [28,198]. Additional detail can be found in the text.

## Data Availability

Not applicable.

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
