# Peer review of "Hepatitis B Virus Epsilon (ε) RNA Element: Dynamic Regulator of Viral Replication and Attractive Therapeutic Target"

_viruses, 2023, doi:10.3390/v15091913_

Round 1

Reviewer 1 Report

Abstract: The abstract does not provide a clear summary of the shortcomings of the article. It only presents an overview of the topic and the goals of the review without highlighting any specific limitations.

Introduction: The introduction provides a general background of HBV and its current treatment strategies but does not clearly state the shortcomings of existing therapies. It mentions the need for alternative therapeutic strategies without providing specific details about the limitations of interferon (IFN)-α and nucleos(t)ide RT inhibitors (NRTIs).

HBV Replication: Molecular Mechanisms and Critical Interactions: This section provides an overview of HBV replication but does not explicitly mention the shortcomings or limitations of the molecular mechanisms and critical interactions discussed. It focuses on describing the steps involved in HBV replication without critically evaluating their effectiveness or highlighting any challenges or gaps in knowledge.

Tackling HBV: Insights into Viral Replication and Evolving Therapeutic Strategies: The section briefly mentions the global burden of HBV infection but does not thoroughly discuss the limitations of current therapeutic strategies. It lacks an in-depth analysis of the challenges faced in developing effective anti-HBV therapies and does not provide a comprehensive overview of evolving therapeutic strategies.

 ε as an Underexploited and Attractive Therapeutic Target:
The reader is not provided with sufficient context to understand the significance of exploring ε as a therapeutic target.
The statement that NRTI and IFN-α therapies are not curative and require lifelong treatment is mentioned without any supporting evidence or references. Including relevant studies or data would strengthen this claim.

Early Characterization of ε: The statement that the ε sequence is highly conserved among other mammalian Hepadnaviruses and different isolates is mentioned without any supporting evidence or references. Providing relevant studies or data to support this claim would add credibility to the statement.

P Protein Structure and Host Interactions: The section acknowledges the absence of structures for HBV or Hepadnaviral P proteins, but it does not critically analyze the impact of this lack of structural information on understanding P protein function and interactions.

 Essential Factors and Dynamic Underpinnings of the ε-P Interaction in HBV Replication: The section briefly outlines the essential factors and regions involved in the ε-P interaction and its downstream functions but does not provide a critical analysis of the experimental evidence supporting these findings.

Structural Dynamics Characterization of Full-Length ε: The mention of preliminary NMR dynamics data and molecular dynamics simulations probing motions on different time scales lacks a critical analysis of the significance or implications of these findings.

Overall, the review article lacks a critical analysis of the shortcomings and limitations of the discussed topics. It provides a general overview of HBV and its replication mechanisms without delving into specific challenges or gaps in knowledge. Additionally, it does not thoroughly evaluate the limitations of current treatment strategies or discuss the potential shortcomings of evolving therapeutic approaches.

Some sentences are long and convoluted, making it difficult to follow the main point. Consider breaking them down into smaller, more concise sentences for better clarity and readability.

Reviewer 2 Report

The review manuscript is well written. It is a comprehensive summary of 175 papers and has sufficient figures. This reviewer enjoyed reading this manuscript. Of note, the 2nd half of this manuscript seems to be a summary of studies published by the authors’ team. So it would be important to assign a paragraph and discuss previous reports which aimed to identify HBV encapsidation inhibitors e.g. Antiviral Res. 2020 Mar;175:104709.

Below find some minor comments:

Line 210: there is a typo in the figure label: “Velmidy”

Line 242: regarding the antiviral function of IFN-alpha, one important paper was not cited- Science. 2014 Mar 14;343(6176):1221-8.

Line 422: DDX3-HBV polymerase interaction was first reported in JVI (J Virol. 2009 Jun;83(11):5815-24)

Line 470: Please specify what “these nucleotides” indicates

Line 492 and Fig. 8D: It is not clear how the percentage of nucleotide identify was calculated – Is the epsilon sequence conserved among HBV with different genotypes or among the family of hepadnaviruses?

Line 497: typo -  Fig. 8D instead of Fig. 7D

Round 2

Reviewer 1 Report

The manuscript has been significantly improved.